# Thermochemical and Toxic Element Behavior during Co-Combustion of Coal and Municipal Sludge

**DOI:** 10.3390/molecules26144170

**Published:** 2021-07-09

**Authors:** Yongchun Chen, Herong Gui, Ziwei Xia, Xing Chen, Liugen Zheng

**Affiliations:** 1National Engineering Laboratory for Protection of Coal Mine Eco-Environment, Huainan 232001, China; hkhj099@163.com; 2National Engineering Research Center of Coal Mine Water Hazard Controlling, Suzhou University, Suzhou 234000, China; hrgui@163.com; 3School of Resource and Environment Engineering, Anhui University, 111 Jiulong Road, Hefei 230601, China; xiaziwei@stu.ahu.edu.cn (Z.X.); 15705610332@163.com (X.C.); 4Anhui Province Engineering Laboratory for Mine Ecological Remediation, Hefei 230601, China

**Keywords:** coal, municipal sludge, co-combustion, thermochemical analysis, toxic elements

## Abstract

The thermochemical and kinetic behavior of co-combustion of coal, municipal sludge (MS) and their blends at different ratios were investigated by thermogravimetric analysis. Simulation experiments were performed in a vacuum tube furnace to determine the conversion behavior of toxic elements. The results show that the combustion processes of the blends of coal and municipal sludge are divided into three stages and the combustion curves of the blends are located between those of individual coal and municipal sludge samples. The DTG_max_ of the sample with 10% sludge addition reaches a maximum at the heating rate of 20 °C/min, indicating that the combustion characteristics of coal can be improved during co-combustion. Strong interactions were observed between coal and municipal sludge during the co-combustion. The volatilization rates of toxic elements decrease with an increasing proportion of sludge in the blends during co-combustion, which indicates that the co-combustion of coal and sludge can effectively reduce the volatilization rate of toxic elements. The study reflects the potential of municipal sludge as a blended fuel and the environmental effects of co-combustion of coal and municipal sludge.

## Highlights

(1)The thermal reactivity of co-combustion of coal can be improved by adding municipal sludge.(2)The kinetic properties and optimum conditions were obtained.(3)The volatilization rate of toxic elements in burned samples was reduced by adding municipal sludge.

## 1. Introduction

In recent years, the global demand for renewable energy has been increasing due to environmental problems and the reduction of fossil fuels. In terms of environmental protection and energy, people who study thermal energy are paying increasing attention to sludge with a certain calorific value and availability; such sludge is mainly derived from the chemical, papermaking and sewage treatment industries. With the rapid development of China’s economy, the discharge of sewage has been increasing. According to the China Urban Construction Statistical Yearbook, the dry sludge production of China’s sewage treatment plants in 2016 was 7,792,232 tons, and there are more than four million tons (dry weight) of municipal sludge produced annually in China [1]. This large amount of sludge is in urgent need of treatment. Sludge has many toxic substances such as pathogens, heavy metals and some organic contaminants, which can cause serious environmental pollution [2]. Moreover, sludge treatment usually consists of thickening, conditioning, dewatering, stabilization and drying, which are performed with various physical, chemical and biological technologies [3]. Therefore, sludge has always been a substantial challenge, and proper management, handling and treatment of sludge are essential.

The utilization of municipal sludge as a renewable energy resource is a feasible way to manage the continuously increasing waste sludge generation. Sludge incineration is an attractive option because it minimizes odor, significantly reduces the volume of the starting material and thermally destroys organic and toxic components in the sludge [4]. Therefore, many countries strongly support the coincineration of sludge as a supplementary fuel in coal-fired power plants, cement kilns, and brick kilns [5]. However, sludge is characterized by a high moisture content, high ash content, high viscosity and low calorific value, which have a negative impact on its incineration [6]. Some researchers have found that this issue can be addressed by adding coal for co-combustion [7,8,9]. Coal is the most important major energy source for power generation, accounting for 36% of global power generation. Its significance varies from region to region: in the countries with large resource endowments, such as China or South Africa, the share of hard coal power generation is more than 80% [10]. China is a country with abundant coal resources, less oil and less natural gas [11]. The main use of coal is direct combustion, which results in a low efficiency of coal resource utilization and serious environmental pollution. The advantages of co-combustion of fuel and waste materials include the reduction of CO_2_ emissions from fossil fuels, the more efficient disposal of waste materials in terms of environmental protection equipment, and elimination of the need for waste incinerators [12]. During the co-combustion process, a number of volatile compounds in sludge are released and combusted in the early stage of co-combustion [13].

The ignition characteristics, combustion characteristics and burnout characteristics of coal and sludge are quite different, and the combustion characteristics of blends are different from those of individual samples. In contrast to the combustion of solid fuels such as coal, the combustion of carbon from sewage sludge requires fewer materials than the combustion of degassed and separated volatile substances. This difference may be attributed not only to the slow thermal decomposition of these wastes, but also to the small amount of carbon in the sludge (approximately 10%). Current combustion systems are determined by the combustion characteristics and ash yield of the feedstocks. The combustion behavior is far different from that of coal because of the large differences in volatile matter, fixed carbon, mineral composition and ash components [14]. There have been many investigations into the thermochemical characteristics of coal and sludge during combustion [15,16]. The co-combustion characteristics of coal and sludge are generally the result of interaction between sludge and coal samples. The ignition and combustion characteristics of coal and sludge blends vary with the proportion of sludge in the blend. Therefore, it is necessary to study different proportions of sludge and the combustion characteristics of the resulting blends. The optimal mixing ratio is particularly important. Therefore, to investigate the combustion characteristics in current real combustion systems, the thermochemical behaviors of coal/sludge blends should be studied. To date, considerable attention has been paid to the co-combustion of coal with sewage sludge [17]; however, the co-combustion of coal and municipal sludge has not been extensively studied [18]. In addition, most of the methods used to study sludge combustion kinetics have directly assumed first-order reactions. In fact, the combustion process of sludge is complicated, and simple assumptions often mask the reaction mechanism. Therefore, the kinetic parameters of combustion were calculated according to the hypothesized models of different reaction mechanisms, and the results were helpful for further understanding the co-combustion process in the combustion of coal and sludge.

Furthermore, the environmental problems caused by sludge incineration are also a problem that cannot be ignored. The existence of toxic elements makes the development of sludge incineration technology challenging. Large amounts of toxic elements may lead to undesirable gas emissions and create substantial potential environmental risks. The retention characteristics of seven elements are discussed and divided into three categories in this paper. As and Pb are evaporated at medium temperatures, mainly through a vaporization-condensation mechanism, leading to their enrichment in fly ash. Co, Cr and Mn are nonvolatile and are evenly distributed in bottom ash and fly ash [19]. Th and U belong to the actinides. The toxicity and radiation of actinides (especially due to the inhalation of alpha radiators) cause greater harm than the other elements discussed here. There is a consensus that the volatilized or particulate toxic elements can travel long distances with wind and result in potential environmental and health impacts [14]. Therefore, with the wide application of sludge incineration technology, research on corresponding pollutant discharge and treatment technologies is becoming increasingly urgent.

In this study, municipal sludge (MS) was selected as the biomass fuel. Due to their enormous availability and cost-effectiveness, the biomass fuel has enormous potential in cogeneration systems for power generation [15,16]. Therefore, the main purpose is to determine: (1) the thermochemical characteristics of bituminous coal, municipal sludge and their blends by thermogravimetric analysis; (2) the optimum mixing ratio and heating rate during co-combustion; (3) related environmental problems.

## 2. Materials and Methods

### 2.1. Ultimate and Proximate Analysis and Samples

The bituminous coal (C) was collected from a coal-fired power plant in Linhuan, Huaibei, China. To represent the chemical characteristics, three groups of furnace coal samples were collected. Municipal sludge was sampled from a sewage treatment plant in Huaibei. The collected coal and municipal sludge samples were air dried, pulverized, passed through a 100-mesh sieve and kept in sealed plastic bags for subsequent analysis. The samples were tested for carbon, hydrogen, oxygen, nitrogen, and sulfur using a Vario EL-3 Vario Macro Cube elemental analyzer (Elementar, Germany) and a WS-S101 automatic sulfur analyzer (Thermofisher, Germany) according to GB/T214-2007. The moisture, ash, volatiles, fixed carbon content, and calorific value of the samples were determined using an SDTGA5000a industrial analyzer (ACTech, Australia) according to GB/T212-2008. The ash composition of the sample was analyzed using a ZAX Primus II X-ray fluorescence spectrometer (Thermofisher, Germany) according to GB/T1574-2007. A series of coal-sludge blends were prepared with mass percentages of sludge of 10%, 20%, 30%, 40% and 50%, which were named 90C10S, 80C20S, 70C30S, 60C40S and 50C50S, respectively.

### 2.2. Thermogravimetric Analysis

The thermochemical characteristics of the coal, municipal sludge and blends were determined by using a the STA449F3 Simultaneous Thermal Analysis system with a temperature precision of ±3 °C. The calorimetric sensitivity and the accuracy of the measured temperature were 0.1 μW and 0.1 °C, respectively. To avoid mass and heat transfer interference, the samples were analyzed under a 100 mL/min air flow from room temperature to 900 °C. For each sample, three different heating rates (10 °C/min, 20 °C/min, 60 °C/min) were used to conduct separate dynamic runs. Three runs were carried out for each sample in order to confirm reproducibility, and the results revealed that the reproducibility was good with standard errors that were within ±3 °C.

### 2.3. Kinetic Analysis

The kinetic characteristics (activation energy and pre-exponential factor) of solid reactions can be determined by both differential and integral methods [20]. The kinetic properties of the coal, municipal sludge and blends thereof were evaluated by the integral method. The fundamental rate equation adopted in all kinetic studied is described in Equation (1):(1)$$\frac{\mathrm{dx}}{\mathrm{dt}}=K\left(T\right)f\left(x\right)$$
where x is the mass conversion rate in the combustion process; f(x) is a hypothetical model, which is determined by the reaction mechanism; t and T represent the combustion time (min) and absolute temperature (K), respectively; and K(T) is the reaction rate, which can be determined by the Arrhenius equation (Equation (2)):(2)$$K\left(T\right)=A\;\exp\left(-\frac{E}{\mathrm{RT}}\right)$$
where A, E and R refer to the precursor (min^−1^), activation energy (kJ/mol) and universal gas constant (8.314 J/(K mol)), respectively. If nonisothermal thermogravimetric analysis is used for combustion, the heating rate (H, K/min) is constant and can be described according to Equation (1) by the following relationship (Equation (3)):(3)$$\frac{\mathrm{dx}}{\mathrm{dT}}=\frac{A}{H}\;\exp(-\frac{E}{\mathrm{RT}})f(x)$$

Integrating both sides of Equation (3) gives Equation (4):(4)$$g\left(x\right)=\int_{0}^{x}\frac{\mathrm{dx}}{f\left(x\right)}=\frac{A}{H}\int_{T_{0}}^{T}\exp(-\frac{E}{\mathrm{RT}})\mathrm{dT}$$
where g(x) is the integral conversion function. Since E/RT >> 1, the right side of Equation (4) can be substituted into Equation (5):(5)$$\frac{A}{H}\int_{T_{0}}^{T}\exp(-\frac{E}{\mathrm{RT}})\mathrm{dT}\;≅\;\frac{{\mathrm{ART}}^{2}}{\mathrm{HE}}(1-\frac{2\mathrm{RT}}{E})\exp(-\frac{E}{\mathrm{RT}})$$

Applying Equation (5) to Equation (4) and taking logarithm of both sides of Equation (4) results in Equation (6):(6)$$\ln\left[\frac{g\left(x\right)}{T^{2}}\right]=\ln\left[\frac{\mathrm{AR}}{\mathrm{HE}}(\frac{1-2\mathrm{RT}}{E})\right]-\frac{E}{\mathrm{RT}}$$

Since the expression ln((1-2RT/E) AR/HE) is in Equation (6) for that most values of E, the temperature range of combustion is basically constant. Therefore, a straight line should be obtained by plotting ln(g(x)/T^2^) against 1/T, and the activation energy E can be calculated from the slope of the line (E/R).

The expressions of g(x) and f(x) for basic kinetic modeling were reported by [21]. The reaction mechanisms are divided into four types of chemical reactions (first, second, third, and n-th order), random nucleation and nuclei growth (two-dimensional and three-dimensional), phase boundary reactions (one-, two-, and three-dimensional), and diffusion (one-, two-, and three-way transport, the Ginstling–Brounshtein equation, and the Zhuravlev equation). The expression of g(x) exhibiting the best correlation (based on R^2^) is considered to represent the model that best demonstrates the kinetic characteristics of mass loss during combustion. To compare the thermochemical properties of the different blends, the kinetic characteristics in this study were estimated by five reaction mechanisms, including chemical reaction (first order), random nucleation and nuclei growth (three dimensional), diffusion (one-way transport), diffusion (three-way transport) and diffusion (Zhuravlev equation). The five reaction mechanisms were used for the combustion of each sample.

### 2.4. Toxic Element Analysis

A BTF-1200C vacuum tube high-temperature furnace (BEO, Hefei, China)was used to simulate the combustion of coal, municipal sludge and blended to determine the retention behavior of toxic elements during co-combustion. The vacuum tube furnace is composed of a control system and furnace. Resistance wire is used as the heating element, which can be heated to the required temperature according to different heating rates. The maximum temperature was 1000 °C, and air was used as the combustion atmosphere. The bottom ash was collected and digested with a 5:5:3 acid solution (HNO_3_: HF: HClO_3_), and the concentrations of Cr, Mn, Co, As, Pb, Th and U in the recovered samples were determined by inductively coupled plasma spectrometry (ICP-MS) according to GBW07406 (GSS-6).

## 3. Results and Discussion

### 3.1. The Coal and Minicipal Sludge Properties

The proximate and ultimate analyses, as well as the toxic element analysis, of the coal and municipal sludge sample, are presented in Table 1. MS had a much higher moisture and ash yield content than the coal, whereas it had lower fixed carbon and calorific values. The oxygen/carbon ratio (O/C) was higher for the sludge than for the coal which meant that sludge had higher hydrophilicity with more polar groups than the coal [22]. In addition, MS contained a higher volatile matter content, volatile fuel ratio [VM/(VM + FC)] and hydrogen carbon ratio (H/C) than the coal. Among these parameters, the higher volatile fuel ratio and hydrogen carbon ratio suggest a lower ignition temperature and more stable flame [23,24].

From the table, the primary chemical components in the coal were SiO_2_ and Al_2_O_3_, whereas those in MS were SiO_2_, Al_2_O_3_, CaO and Fe_2_O_3_. Compared with the coal, elevated contents of alkali and alkali-earth oxides including CaO, K_2_O, MgO and Na_2_O were found in the MS samples. It has been reported that a high concentration of alkali and alkali-earth oxides can lower the melting temperature and create agglomeration problems during combustion [25].

### 3.2. Thermal Behaviors of Coal, Municipal Sludge and Their Blends

The thermodynamic behavior indicated by thermogravimetric analysis can provide useful information related to the flammability and conversion behavior of coal and sludge [26]. Figure 1 presents the thermogravimetric (TG) and derivative thermogravimetric (DTG) curves produced from the temperature programmed combustion of coal, municipal sludge and their blends at heating rates of 10, 20 and 60 °C/min. The combustion characteristic parameters of the combustion curves of coal, municipal sludge and their blends, including the temperature interval, peak temperature (T_max_) and maximum combustion rate (DTG_max_), are listed in Appendix A. As shown in Figure 1, the DTG curves of municipal sludge are quite different from those of the coal. Three stages, namely, dehydration of moisture (stage I), release of (stage II) and char combustion (stage III), are observed during the combustion process of sludge [27].

As shown in Figure 1b,d,e, the DTG curve of the coal is completely different from the DTG curve of the sludge. At heating rates of 20 °C/min, the DTG curves of coal samples have obvious peaks at 529 °C, and the corresponding maximum burning rate (DTG_max_) is 2.478%/°C. The combustion of volatile matter was inconspicuous, and the fixed carbon combustion stage showed significant weight loss peaks in the DTG curve. Sludge contains highly complex organics, such as proteins, cellulose, hemicellulose and lignin, which can be regarded as special sources of bioenergy [28]. The weight loss of sludge corresponds to the decomposition of carbohydrates, proteins and aliphatic compounds [29]. From room temperature to 200 °C, the first stage of MS combustion corresponds to the loss of free water and chemically bound water. The evaporation of water absorbs the latent heat of vaporization and results in an absorption peak in the DTG curve. In the water dehydration stage, the weight loss and maximum burning rate of the MS are both high, which is consistent with the moisture content. The weight loss of MS in the second stage was mainly due to the degassing and combustion of volatiles. At this stage, the DTG curve has a significant weight loss peak and a subsequent side peak. The reason is that the volatile components in the sludge are complicated and the chemical bond strength of each component varies. At the heating rate of 20 °C/min, the DTG curve of MS combustion showed a peak at 293 °C, and the corresponding maximum combustion rate (DTG_max_) was 1.050%/°C. The larger the DTG_max_ is, the easier the fuel burns and the better the combustion characteristics [30]. In the subsequent third stage, the weight loss of the MS between 380 and 600°C is due to the burning of fixed carbon. Due to the low fixed carbon content in the sludge, the weight loss W (%) and the maximum combustion rate DTG_max_ of the MS are lower than that of the coal at this stage. Comparing the TG and DTG curves of coal and MS, it can be clearly seen that the degassing temperature of the volatile matter and the burnout temperature of the combustibles in MS are earlier than those in the coal, which proves that the sludge contains more volatile matter and that the coal contains more fixed carbon. Once the combustible content in the material is exhausted, the weight loss curve of the ash remains stable. The TG curves show that the final residual amount in coal is 42.2%, and the final amount of residue in the sludge is 56.3%. The total weight loss of the sludge is 14.1% lower than that of the coal. Compared with the combustion characteristics of coal in the second stage, the temperature (T_max_) of the maximum combustion rate is lower than the corresponding temperature for the coal, and the DTG_max_ of the coal is much higher than that of the sludge. The different thermal behaviors between coal and sludge can be attributed to differences in volatile matter, carbon content and lignocellulose composition [31]. The combustion of sludge is quite different from the combustion of coal. The combustion of coal mainly depends on fixed carbon, and the combustion of sludge mainly depends on the activity of volatile substances. In comparison with the ignition temperature between coal (about 324 °C) and municipal sludge (134–153 °C), the selected sludge samples are ignited at lower temperature, which may be explained by the higher volatile matter, oxygen content and volatile fuel ratio in sludge. Therefore, it is desirable to enhance the ignition performance by adding municipal sludge during combustion of the coal.

The composition of the coal and sludge, the sludge mass ratio and the heating rate are the main factors affecting the combustion characteristics of coal/sludge blends. The TG curves of blends of coal and sludge with different proportions are shown in Figure 1. All the curves of the blends are between the curves of the individual components, and the contributions of sludge and coal to the blends are evident. The thermal characteristics vary among the selected materials. With increasing of combustion temperature, the degradation of samples is accompanied by a weight loss after the evaporation of water. The ash yields of C/MS blends decrease gradually with the increasing sludge when the temperature rose from room temperature to 528 °C, while the residue mass of blends maintains between 51.7 and 57.9%.

The DTG curves of the blends present three or four peaks related to the dehydration, combustion of volatile matter and char combustion, which are attributed to the dominant effects of the selected samples in blends. It can be found that when the MS is incorporated into coal, the combustion of the blend exhibits a distinct peak in the temperature range of 188–364 °C, which can be explained by the decomposition of low molecular weight volatiles in the municipal sludge. The peak in stage III is attributed to the combustion of fixed carbon in coal and sludge. According to Appendix A, when the temperature rises from room temperature to approximately 400 °C, the ash yield of the blends decreases with increasing of sludge [32]. From 400 °C to the burnout temperature, the peak temperature (T_max_) is advanced with increasing sludge ratio, indicating that decomposition begins at low temperatures. With the elevated additive amount of MS, the DTG_max_ values of Stage II increase from 0.063%/°C to 0.543%/°C, while the DTG_max_ values decrease from 2.587%/°C to 1.578%/°C at the Stage III. At a heating rate of 10, 60 °C/min, DTG_max_ reaches a maximum in the coal sample and gradually decreases with increasing sludge addition amount; the added amount of sludge should not exceed 30%. In the blends, DTG_max_ reaches a maximum in the C90S10 sample, indicating that the combustion performance does not change much when the amount of sludge added is small, which is consistent with the combustion characteristics of coal-only samples. It is worth noting that when the heating rate is 20 °C/min, DTG_max_ reaches a maximum in the C90S10 sample; this value is 0.109% higher than the maximum burning rate when coal is burned alone, which indicates that the sludge addition amount can be effectively increased. It has been reported that the thermal reactivity is positively proportional to DTG_max_ and inversely correlated with the peak temperature [33]. From this relationship, the thermal reactivity of coal can be promoted by adding sludge. Nevertheless, the proportion of sludge in comaterials should be suitable and reasonable due to the different DTG_max_ values at different heating rates.

When considering a co-combustion option, it is important to evaluate the synergistic effects between fuels. Assuming that there is no interaction between these two fuels, the overall mass loss of the blends is the weighted average of the individual values (as calculated by Equation (7).
(7)$$\frac{\mathrm{dm}}{\mathrm{dt}}{=x}_{1}{\left(\frac{\mathrm{dm}}{\mathrm{dt}}\right)}_{c}{+x}_{2}{\left(\frac{\mathrm{dm}}{\mathrm{dt}}\right)}_{s}$$
where (dm/dt)_c_ and (dm/dt)_s_ represent the weight loss rates of coal and sludge, respectively; and x_1_ and x_2_ are the proportions of coal and sludge in the blends, respectively. The experimental and calculated co-combustion TG curves at a heating rate of 10 °C/min with the four sludge ratios are plotted in Figure 2. Significant deviations appear in the experimental and calculated TG curves during the co-combustion. The weight loss of the experimental curve is higher than that of the calculated curve between room temperature and 500 °C, and the opposite is observed above 500 °C. This result suggests that interactions existed between the two components during co-combustion and were more pronounced at higher sludge proportions.

### 3.3. Effect of Heating Rate on Thermal Characteristics

The thermal characteristics at different heating rates are important data for the design and operation of co-combustion applications. When the heating rate is different, there are some differences in the combustion characteristic curves. Figure 1 shows the thermogravimetric curves of coal, sludge and blends at heating rates of 10, 20 and 60 °C/min. Some studies have focused on the TG curves of the coal/sludge blends and indicated that the curves of the blends follow a sequence between the curves of individual samples of coal and sludge [34]. In this study, the DTG curves of the blends were arranged in sequence between individual samples of coal and sludge at a heating rate of 10 °C, but the DTG curves of the blends were not in order at 20 and 60 °C/min. This result occurred because when the heating rate was 10 °C, the heat transfer is slower and the combustion was more uniform.

Figure 3 shows the results for single samples of coal and sludge and the thermogravimetric analysis of C50S50 at different heating rates. The higher the heating rate is, the smaller the weight loss of the sample at the same temperature, because the solid-phase yield becomes larger and the yield of volatile products decreases [35]. As the heating rate increases, the DTG curve moves toward the high-temperature region, the peak value increases, the combustion region becomes wider, and the rate of loss of combustion becomes larger. The wider and lower the DTG curves of the blends are, the higher the T_max_, indicating that the reaction efficiency is low and the combustion cycle is longer. The reason for this result is that the higher the heating rate is, the shorter the time elapsed during pyrolysis, the lower the degree of the reaction, and the lower the influence in the increase of temperature gradient; hence, some pyrolysis products cannot diffuse in time, leading to thermal hysteresis [36]. Moreover, the slower the heating rate is, the more obvious the weight loss and the higher the burnout rate. This is because when the rate of temperature rise is slow, the sample has a greater reaction time, so the weight loss is greater. Among different heating rates, the DTG_max_ of all samples increased at a higher heating rate, indicating an increase in the combustion performance of the sample. The DTG_max_ of coal and sludge increased by 2.318%/°C and 2.017%/°C, respectively, at a heating rate of 60 °C/min compared with that of 10 °C. The DTG_max_ of the blends 90C10S, 80C20S, 70C30S, 60C40S and 50C50S increased by 1.086%/°C, 0.872%/°C, 1.760%/°C, 1.326%/°C and 1.132%/°C, respectively, at a heating rate of 60 °C/min compared with that of 10 °C. It can be seen that in the pyrolysis process of coal and sludge, reducing the heating rate can make the sludge decompose more thoroughly, which is beneficial to the reduction of sample volume during incineration. However, when the heating rate is low, the pyrolysis rate and heat release rate of coal and sludge pyrolysis are small, which is not conducive to rapid processing and release of the samples.

### 3.4. Kinetic Parameters of Coal, Sludge and Their Blends

The results of kinetic analysis of the raw materials and their blends calculated by the five reaction models are shown in Table 2, where R^2^ is the correlation coefficient. The high coefficient values indicate that the five reaction models fit these processes satisfactorily. The activation energies (E) of the blends in stage II increased with increasing sludge percentage in the blends, while the activation energies of stage III showed the opposite trend. The trend in activation energy can be attributed to the fact that the pyrolysis of sludge occurs mainly in stage II, while the pyrolysis of coal occurs in stage III. The tendencies and values in this study are partially consistent with relevant previous studies [37].

Activation energy, as a potential barrier, provides important data on the minimum energy required to support a reaction, and has been the focus of intensive research [38]. The proportion of sludge added in the blends can reduce the activation energy in stage III, and, therefore, MS can be considered a suitable clean cofuel; however, the proportion of sludge added should be reasonable because of the increases in activation energy in stage II. Therefore, the kinetic behavior of coal could be improved by the suitable addition of sludge.

In addition, the base/acid ratio of blends should be less than 0.5 to ensure flame stability and prevent fouling [39]. It was determined that the base/acid ratio of all the blends was less than 0.5. Combining the aforementioned results, the additive proportion of 10% appears to be the optimum blend ratio for MS, and could be combustion directly in the boiler systems as used for coal.

### 3.5. Retention Behavior of Toxic Elements during Co-Combustion of Coal with Sludge

The utilization of coal and sludge as co-fuels has both positive and negative environmental impacts. Despite the benefits of this approach in dealing with excess coal and sludge, it introduces potential environmental impacts due to emission of trace elements (especially environmentally sensitive elements) during combustion [40]. Due to the high ash content of sludge, slagging and scaling may occur on the heating surface of the boiler. All of these problems must be considered to successfully apply co-combustion technology to recover energy from sludge. Based on these requirements, a sludge addition of less than 50% is considered appropriate. Therefore, 90:10, 80:20, 70:30, 60:40 and 50:50 coal/sludge blends were applied to study the retention characteristics of toxic elements (As, Pb, Co, Cr, Mn, Th and U).

Retention percentages of toxic elements in burned coal, sludge and blends are presented in Figure 4. The retention percentages of toxic elements are derived from the radio of the content of toxic elements in the sample after combustion and the content of toxic elements in the sample. The potential release of toxic elements into the atmosphere can be inferred from measurements of the retention of individual elements in the blends. The average retention of toxic elements in sludge was 20.21%, 21.76% and 18.11% at heating rates of 10, 20 and 60 °C/min, while that in coal was 60.14%, 58.01% and 62.40%, respectively. The volatilization rate of toxic elements in sludge combustion products is less than that of coal, indicating that the combustion of sludge is less harmful to the environment than the combustion of coal. By adding different proportions of sludge, the retention percentages of toxic elements in the samples increases, indicating that the volatilization rate is reduced. The more sludge is added, the lower the volatilization rate of toxic elements.

The average value of each element was calculated, and the volatilization rates of the toxic elements in the combustion products from high to low followed the order Pb, Mn, Cr, Co, U, As and Th. Pb is a relatively volatile heavy metal that generates PbCl_2_ during combustion and is volatilized with flue gas and fly ash. Mn, Cr and Co are all poorly volatile elements, and most of them are enriched in the slag [41]. Cr is not volatile at 800–1500 °C and is mainly present in the slag in the form of FeCr_2_O_4_ and CaO•Cr_2_O_3_. When power plants co-combust sludge with coal, the resultant slag and fly ashes consist mainly of oxides such as SiO_2_, Al_2_O_3_, CaO, Fe_2_O_3_, and P_2_O_5_. Such oxides act as carriers in different forms for the adsorption of As, such as Ca(AsO_3_)_2_, Ca_3_(AsO_4_)_2_, and FeAsO_4_ [42]. The As volatilization rules of different chemical forms are different, with 1000 °C as the cut-off point. When the temperature is lower than 1000 °C, the volatilization rate of As increases with increasing temperature and time, but at temperatures higher than 1000 °C, the volatilization rate decreases with increasing temperature.The volatilization rate of As is small in this paper because increased temperature suppresses the volatilization of As. Since As easily reacts with CaO, the higher the temperature is, the more easily gaseous As reacts with CaO to form poorly volatile compounds such as Ca(AsO_4_), that remain in the slag [43]. The volatility of Pb and Mn in the samples is higher (greater than 70%), while the volatility of Th and U is lower. Therefore, the co-combustion of coal and sludge can reduce the emission of toxic elements and can be used for environmental protection.

## 4. Conclusions

Three stages, namely, dehydration of moisture (stage I), release of (stage II) and char combustion (stage III), are observed during the combustion process of sludge. The combustion process of coal has only one stage of char combustion. As the blending of municipal sludge with coal is increased, the DTG_max_ increases and the activation energy decreases in the main combustion regions, indicating that the thermal reactivity and kinetic behavior are enhanced. The additive proportion of 10% appears to be the optimum blend ratio for municipal sludge. The higher the rate of temperature rise is, the greater the rate of combustion of samples, but the decomposition of samples is not complete. The addition of an appropriate municipal sludge amount can reduce the volatilization rate of toxic elements in coal. The greater the added sludge content is, the lower the volatilization rate of the samples, so the co-combustion of coal and municipal sludge can be used for environmental protection.

## Figures and Tables

**Figure 1 molecules-26-04170-f001:**
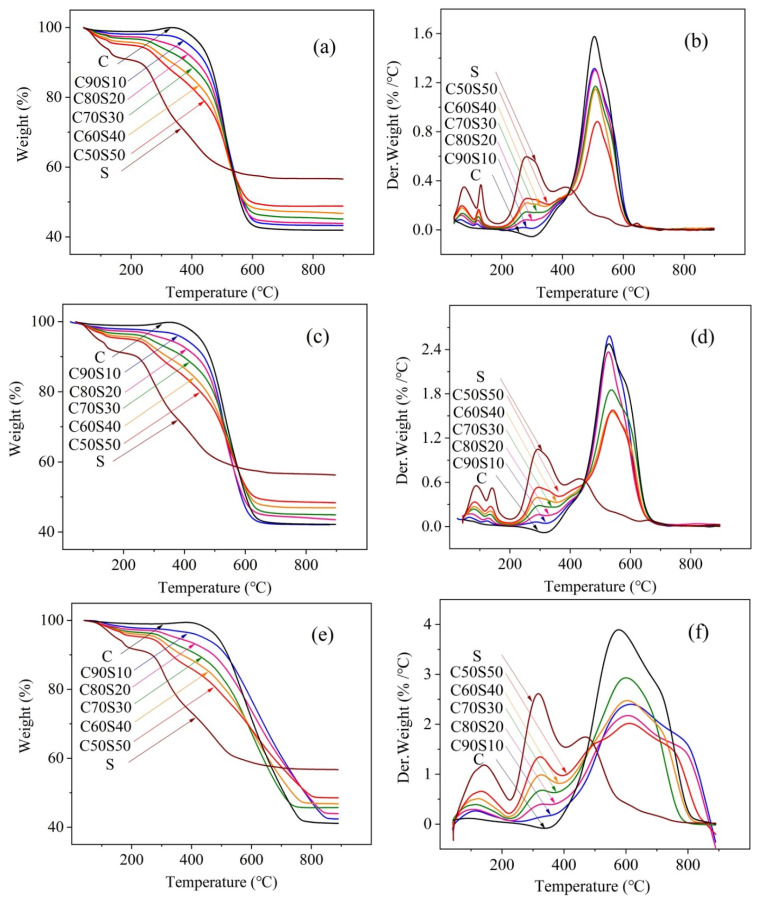
The TG and DTG curves of coal, municipal sludge and blends thereof at heating rates of 10 (**a**,**b**), 20 (**c**,**d**) and 60 (**e**,**f**) °C/min.

**Figure 2 molecules-26-04170-f002:**
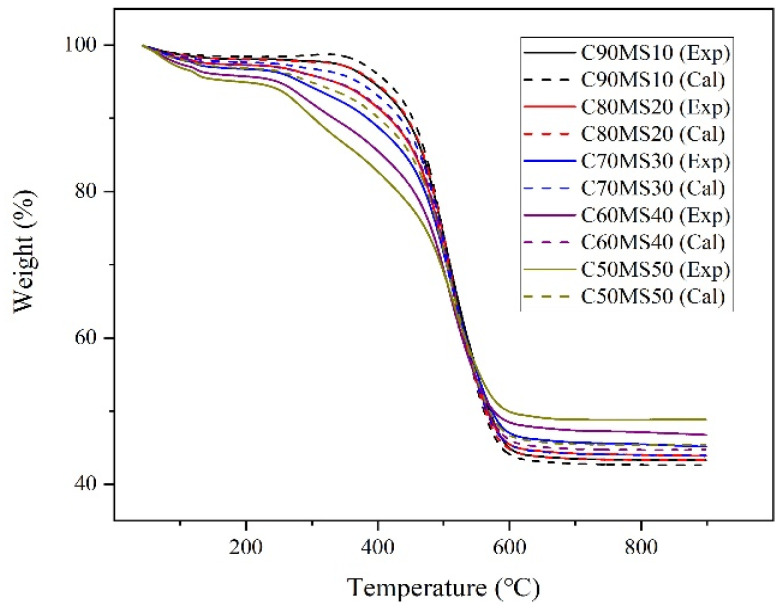
Comparison of experimental and calculated TG curves for five blend ratios of coal/municipal sludge blends at a heating rate of 10 °C/min. Exp, experimental curve; Cal, calculated curve.

**Figure 3 molecules-26-04170-f003:**
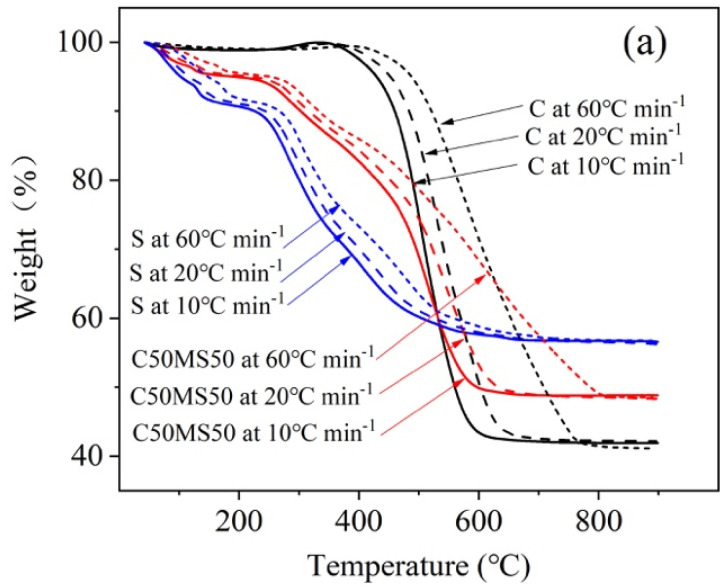
TG (**a**) and DTG (**b**) curves of coal, municipal sludge and C50S50 at different heating rates of 10, 20 and 60 °C/min.

**Figure 4 molecules-26-04170-f004:**
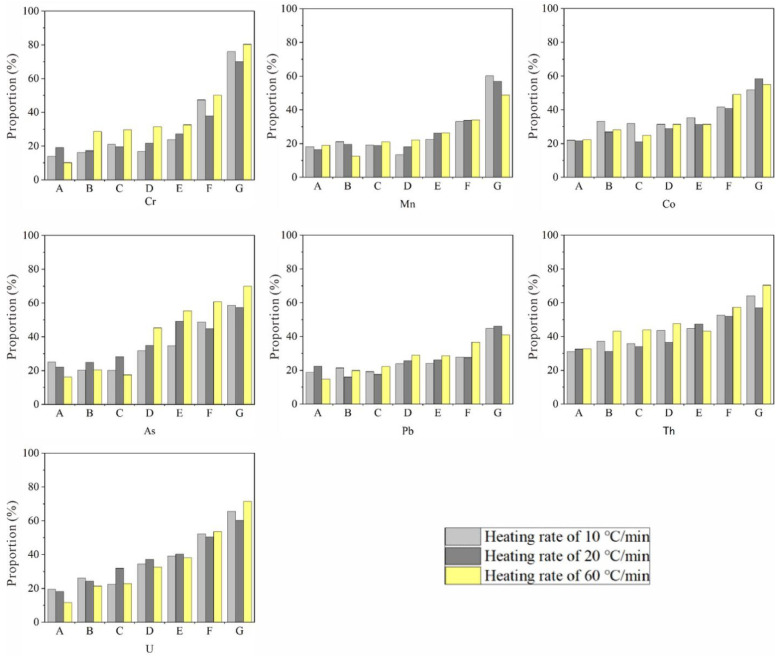
The percentage of toxic elements retained in the combustion of of coal, municipal sludge and their blends. A: coal, B: C90MS10, C: C80MS20, D: C70MS30, E: C60MS40, F: C50MS50, G: municipal sludge.

**Table 1 molecules-26-04170-t001:** The physic-chemical characterizations of the coal (C) and municipal sludge (MS).

**Proximate analysis, db (wt.%)**	**M**	**A**	**VM**	**FC**	**LHV (MJ/kg)**		
C	1.09	38.7	21.4	38.8	19.8		
MS	8.02	50.2	39.6	2.16	11.5		
**Ultimate analysis, db (wt.%)**	**C**	**H**	**O**	**N**	**S**		
C	63.8	4.20	10.5	0.93	0.84		
MS	22.9	3.38	17.1	3.04	1.11		
**Ash analysis (wt.%)**	**SiO_2_**	**Al_2_O_3_**	**CaO**	**Fe_2_O_3_**	**K_2_O**	**Na_2_O**	**MgO**
C	18.5	10.2	0.35	1.56	0.49	0.14	0.19
MS	32.8	9.42	7.76	4.48	1.36	0.56	1.29
**Trace element (mg/kg)**	**Cr**	**Mn**	**Co**	**As**	**Pb**	**Th**	**U**
C	14.7	11.1	1.07	2.08	1.64	2.15	2.70
MS	1.70	0.71	0.05	0.06	0.03	0.22	0.04

Notes: db, dry basis; M, moisture; A, ash yield; VM, volatile matter; FC, fixed carbon; HV, heating value.

**Table 2 molecules-26-04170-t002:** Kinetic properties of coal, municipal sludge and their blends at heating rates of 10 (**a**), 20 (**b**) and 60 (**c**) °C/min.

(**a**)				
**Sample**	**Stage II**		**Stage III**	
	**E (KJ mol^−1^)**	**R^2^ (%)**	**E (KJ mol^−1^)**	**R^2^ (%)**
C			79.80	99.70
C90MS10	4.662	97.99	65.96	99.65
C80MS20	38.78	99.50	57.24	99.32
C70MS30	45.86	99.63	55.82	99.45
C60MS40	48.05	99.55	55.44	98.68
C50MS50	52.94	99.41	29.56	99.43
MS	41.45	99.82	58.51	98.36
(**b**)				
**Sample**	**Stage II**		**Stage III**	
	**E (KJ mol^−1^)**	**R^2^ (%)**	**E (KJ mol^−1^)**	**R^2^ (%)**
C			80.05	99.63
C90MS10	33.30	98.24	72.21	99.50
C80MS20	43.43	99.73	58.21	99.30
C70MS30	46.37	99.57	58.03	99.44
C60MS40	47.68	99.82	56.88	99.49
C50MS50	48.23	99.77	56.00	99.51
MS	46.09	99.84	54.53	98.06
(**c**)				
**Sample**	**Stage II**		**Stage III**	
	**E (KJ mol^−1^)**	**R^2^ (%)**	**E (KJ mol^−1^)**	**R^2^ (%)**
C			87.07	99.44
C90MS10			61.26	99.27
C80MS20	51.33	99.61	57.77	99.63
C70MS30	51.80	99.86	55.39	99.65
C60MS40	52.71	99.69	54.71	99.58
C50MS50	63.98	99.90	42.91	99.41
MS	52.63	99.44	61.21	98.68

Notes: E, activation energy; R^2^, correlation coefficient.

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
