# Peer review of "Thermochemical and Toxic Element Behavior during Co-Combustion of Coal and Municipal Sludge"

_molecules, 2021, doi:10.3390/molecules26144170_

Round 1
Reviewer 1 Report
The work carried out on this paper is extensive, with a large number of analyzes and comments on the results. The topic is also interesting because the main objective is to manage a complicated waste to manage municipal sludge. On the other hand, once I have reviewed all the work, I think that Molecules is not the appropriate journal for this topic, but that Energies or Processes would have been more successful in choosing the journal.
Regarding the specific comments of the review:
- The title appears twice the word "behavior", I think it could be changed to make it more appropriate.
- line 58, "high density", I don't think this is a characteristic of municipal sludge, as it has a high degree of humidity.
- line 67-70, I don't know why we are talking about pyrolysis here, without making a brief introduction of the reason. I would delete it. By the way, the solid-phase pyrolysis is not "carbon", but char.
- line 121, the title has to be "2. Materials and methods"
- line 122, the title could be "2.1. Ultimate and proximate analysis and samples"
- In section 2. Materials and Methods does not appear how the equipment to obtain calorific value.
- In section 2. What is the air flow in thermogravimetric analysis?
- Line140. "900 degrees" NO, better "900ºC".
- In section 2.4. It is necessary to put how the toxic retention calculation shown in Figure 4 is performed.
- Line 186, "1000 degrees", better "1000ºC"
- Line 191. Could get better "3.1. The coal and minicipal sludge properties"
- Table 1. Is moisture on dry basis too?
- Table 1 HV (MJ / Kg) what is LHV or HHV?
- Table 1 There is no ultimate and proximate analysis of blends?
- Line 202: daf is dry and ash free, missing "ash"
- Table 1, Why ultimate analysis is on dry ash free and not dry basis
- Figure 1. title put a, b, .... f, to which heating rates correspond
- Figure 1. b, d and f, vertical axis, units "(% / ºC-1)" better this way (% / ºC)
- Table 2, has the same information as Figure 1, I would not put it on the paper, maybe it is better that it is in complementary material.
- Table 2, There are three tables, in the title define a, b and c for each of the heating rates.
- lines 242, 252 "the precipitation of volatiles" is not an adequate expression.
- line 281. "release of volatile matter", I think better that it is "combustion of volatile matter", the flow is air and is not an inert gas.
- Figure 2. It is too small, you have to make it bigger
- Figure 3. Why have you chosen C50S50 and not another of the mixes, what is the reason?
- Table 3. There are three tables, in the title define a, b and c for each of the heating rates
- line 381. Which means the base / acid ratio, this has not been defined before.
- line 397. "the retention ratios", is better retention percentages.
- line 400. Where do these values ​​for percentages of total toxic elements come from?
- In Section 3.5. It would be good to know what the maximum levels of these metals are in the environmental law, since if initially these metals are at very, very low levels with respect to the maximum allowed level, this section would not make much sense.
- Figure 4, put the most appropriate title, it is the percentage of toxic retained in the combustion of .....
- Figure 4. The Y-axis is missing the title, "%"
Author Response
Reviewer #1
- The title appears twice the word "behavior", I think it could be changed to make it more appropriate.
RE: The title is revised to ‘Thermochemical and toxic element behavior during co-combustion of coal and municipal sludge’ in the resubmitted manuscript.
- line 58, "high density", I don't think this is a characteristic of municipal sludge, as it has a high degree of humidity.
RE: The original sentence has been changed to “…high ash content, high viscosity and low calorific value…”
- line 67-70, I don't know why we are talking about pyrolysis here, without making a brief introduction of the reason. I would delete it. By the way, the solid-phase pyrolysis is not "carbon", but char.
RE: We deleted the words about the pyrolysis.
- line 121, the title has to be "2. Materials and methods"
RE: The original title has been changed to "2. Materials and methods".
- line 122, the title could be "2.1. Ultimate and proximate analysis and samples"
RE: The original title has been changed to "2.1. Ultimate and proximate analysis and samples".
- In section 2. Materials and Methods does not appear how the equipment to obtain calorific value.
RE: The moisture, ash, volatiles, fixed carbon content, and calorific value of the samples were determined using an SDTGA5000a industrial analyzer.
- In section 2. What is the air flow in thermogravimetric analysis?
RE: To avoid the mass and heat transfer interference, the samples were analyzed under a 100ml/min air flow from room temperature to 900 °C.
- Line140. "900 degrees" NO, better "900ºC".
RE: The original sentence has been changed to “…900 °C”.
- In section 2.4. It is necessary to put how the toxic retention calculation shown in Figure 4 is performed.
RE: The retention percentages of toxic elements are derived from the radio of the content of toxic elements in the sample after combustion and the content of toxic elements in the sample.
- Line 186, "1000 degrees", better "1000ºC"
RE: The original sentence has been changed to “…1000 °C”.
- Line 191. Could get better "3.1. The coal and minicipal sludge properties"
RE: The original title has been changed to "3.1. The coal and minicipal sludge properties".
- Table 1. Is moisture on dry basis too?
RE: The moisture, ash, volatiles, and fixed carbon content of the samples on dry basis were determined using an SDTGA5000a industrial analyzer.
- Table 1 HV (MJ / Kg) what is LHV or HHV?
RE: Lower heating value of samples were analyzed in this paper.
- Table 1 There is no ultimate and proximate analysis of blends?
RE: The ultimate and proximate characteristics of blends were calculated by the blending ratios of individual feedstock.
- Line 202: daf is dry and ash free, missing "ash"
RE: These mistakes have been corrected.
- Table 1, Why ultimate analysis is on dry ash free and not dry basis
RE: the ultimate is on dry basis, the mistake was revised.
- Figure 1. title put a, b, .... f, to which heating rates correspond
RE: The original title has been changed to “Fig. 1. The TG and DTG curves of coal, municipal sludge and blends thereof at heating rates of 10(a, b), 20(c, d) and 60(e, f) °C/min.”
- Figure 1. b, d and f, vertical axis, units "(% / ºC-1)" better this way (% / ºC)
RE: These mistakes have been corrected. Detailed modifies have been made in the Fig. 1.
- Table 2, has the same information as Figure 1, I would not put it on the paper, maybe it is better that it is in complementary material.
RE: I have put table 2 on the complementary material.
- Table 2, There are three tables, in the title define a, b and c for each of the heating rates.
RE: These mistakes have been corrected. The original title has been changed to “Table S1 Combustion characteristics for coal, municipal sludge and their blends at heating rate of 10 (a), 20 (b) and 60 (c) °C/min.”
- lines 242, 252 "the precipitation of volatiles" is not an adequate expression.
RE: ‘the precipitation of volatiles’ was revised to ‘the combustion of volatiles’.
- line 281. "release of volatile matter", I think better that it is "combustion of volatile matter", the flow is air and is not an inert gas.
RE: The original sentence has been changed to “…combustion of volatile matter…”.
- Figure 2. It is too small, you have to make it bigger
RE: Detailed modifies have been made in the Fig. 2.
- Figure 3. Why have you chosen C50S50 and not another of the mixes, what is the reason?
RE: Due to the high ash content of sludge, slagging and scaling may occur on the heating surface of the boiler. All of these problems must be considered to successfully apply co-combustion technology to recover energy from sludge. Based on these requirements, a sludge addition of less than 50 % is considered appropriate.
- Table 3. There are three tables, in the title define a, b and c for each of the heating rates
RE: These mistakes have been corrected. The original title has been changed to “Table 2 Kinetic properties of coal, municipal sludge and their blends at heating rates of 10 (a), 20 (b) and 60 (c) °C/min.”
- line 381. Which means the, base / acid ratio this has not been defined before.
RE:Tthe base/acid ratio was defined in the resubmitted manuscript.
- line 397. "the retention ratios", is better retention percentages.
RE: The original sentence has been changed to “Retention percentages…”.
- line 400. Where do these values ​​for percentages of total toxic elements come from?
RE:The retention percentages of toxic elements are derived from the radio of the content of toxic elements in the sample after combustion than the content of toxic elements in the sample.
- In Section 3.5. It would be good to know what the maximum levels of these metals are in the environmental law, since if initially these metals are at very, very low levels with respect to the maximum allowed level, this section would not make much sense.
RE: The sentence was revised to describe the transformation behavior of toxic element during co-combustion.
- Figure 4, put the most appropriate title, it is the percentage of toxic retained in the combustion of .....
RE: The original title has been changed to “Fig. 4. The percentage of toxic retained in the combustion of of coal, municipal sludge and their blends.”
- Figure 4. The Y-axis is missing the title, "%"
RE: These mistakes have been corrected. Detailed modifies have been made in the Fig. 4.

Reviewer 2 Report
The manuscript is an clearly written and needs some more details in different parts. After major revisions the manuscript can be accepted for publication.
- Introduction: Several statements in the introduction shall be verified by literature if they are not a general/simple relationship. For example see statements in L. 103-104, 108-109 and a number of other sentences.
- L. 44-46: What are the yearly masses of the different sludges apart from municipal sludge?
- L. 69: Please clarify the “problem of energy self-sufficiency”.
- L. 77-79: What is the statement of this sentence?
- Methods: Please add the standards used for the analyses.
- L. 168: Sentences ends unexpectedly.
- L. 181: Why vacuum?
- L. 193-194: Such statement is valid only on dry basis. Both fuels only have nearly same values for ash and moisture.
- Table 1: Are the ash analyses absolute or relative values? Which further oxides are in the ashes? For C the SO3 is not counted and for MS at least P2O5 is important. For the composition of sludge and the ash please add some statement in the text if this a typical composition (for example see some papers by S. V. Vassilev from years 2010-2015 and others). Please check the contents of heavy metals which are appearing wrongly for MS vs. C.
- L. 242, 252: The word “precipitation” feels wrong in the context. Instead “degassing” or others would be better.
- L. 315-317: This positive effect for fuel blends was found in other studies too (for example Fuel, 239 (2019) 1194-1203). Is this caused by the ash here (see comment on Table 1)?
- L. 337-340: Is the response of the fuel delayed related to the oven?
- Figure 3: Caption of y axis of DTG is wrong.
- Figure 4: Caption of y axis is missing.
- Part 3.5: Is residual char in the sample after buring responsible for retention?
- Conclusion: A little bit more details would be good for the readers.
- Units: Temperatures shall be °C and not degrees (L. 140, 185/186 and so on) and some mistakes in other units.
Author Response
- Introduction: Several statements in the introduction shall be verified by literature if they are not a general/simple re lationship. For example see statements in L. 103-104, 108-109 and a number of other sentences.
RE: The references were added in the revised manuscript.
- L. 69: Please clarify the “problem of energy self-sufficiency”.
RE: We deleted the words about the pyrolysis.
- L. 77-79: What is the statement of this sentence?
RE: The sentence was revised to ‘Current combustion systems are determined by the combustion characteristics and ash yield of the feedstocks’.
- Methods: Please add the standards used for the analyses.
RE: The samples were tested for carbon, hydrogen, oxygen, nitrogen, and sulfur using a Vario EL-3 Vario Macro Cube elemental analyzer and a WS-S101 automatic sulfur analyzer according to GB/T214-2007. The moisture, ash, volatiles, fixed carbon content, and calorific value of the samples were determined using an SDTGA5000a industrial analyzer according to GB/T212-2008. The ash composition of the sample was analyzed using a ZAX Primus II X-ray fluorescence spectrometer according to GB/T1574-2007. The concentrations of Cr, Mn, Co, As, Pb, Th and U in the recovered samples were determined by inductively coupled plasma spectrometry (ICP-MS) according to GBW07406 (GSS-6).
- L. 168: Sentences ends unexpectedly.
RE: We added the sentence “The five reaction mechanisms were used for the combustion of each sample.”
- L. 181: Why vacuum?
RE: The products put into the furnace will not change the original use properties because of the reaction with the air when it is in a vacuum state.
- L. 193-194: Such statement is valid only on dry basis. Both fuels only have nearly same values for ash and moisture.
RE: The mistakes were revised in the manuscript.
- Table 1: Are the ash analyses absolute or relative values? Which further oxides are in the ashes? For C the SO3 is not counted and for MS at least P2O5 is important. For the composition of sludge and the ash please add some statement in the text if this a typical composition (for example see some papers by S. V. Vassilev from years 2010-2015 and others). Please check the contents of heavy metals which are appearing wrongly for MS vs. C.
RE: The contents of heavy metals were checked and revised in the resubmitted manuscript.
- L. 242, 252: The word “precipitation” feels wrong in the context. Instead “degassing” or others would be better.
RE: The original words have been changed to “degassing”.
- L. 315-317: This positive effect for fuel blends was found in other studies too (for example Fuel, 239 (2019) 1194-1203). Is this caused by the ash here (see comment on Table 1)?
RE: yes
- Figure 3: Caption of y axis of DTG is wrong.
RE: This mistake has been corrected.
- Figure 4: Caption of y axis is missing.
RE: These mistakes have been corrected.
- Part 3.5: Is residual char in the sample after buring responsible for retention?
RE: yes
- Conclusion: A little bit more details would be good for the readers.
RE: Detailed modifies have been made in the Conclusion.
- Units: Temperatures shall be °C and not degrees (L. 140, 185/186 and so on) and some mistakes in other units.
RE: These mistakes have been corrected.
